# Blood Clot Dynamics and Fibrinolysis Impairment in Cancer: The Role of Plasma Histones and DNA

**DOI:** 10.3390/cancers16050928

**Published:** 2024-02-25

**Authors:** Matti Ullah, Shahsoltan Mirshahi, Azadeh Valinattaj Omran, Iman Aldybiat, Sullyvan Crepaux, Jeannette Soria, Geneviève Contant, Marc Pocard, Massoud Mirshahi

**Affiliations:** 1CAP-Paris Tech., INSERM U1275, Université Paris Cité, Hôpital Lariboisière, 75010 Paris, France; mattiullah@hamdard.edu.pk (M.U.); azadeh.valinattajomran@lspm.cnrs.fr (A.V.O.); iman.aldybiat@inserm.fr (I.A.); sullyvan.crepaux@stago.com (S.C.); jeannette.soria@inserm.fr (J.S.); marc.pocard@inserm.fr (M.P.); 2Faculty of Pharmacy, Hamdard University, Islamabad Campus, Islamabad 45550, Pakistan; 3Prospective Research, Diagnostica Stago, 92230 Gennevilliers, France; shahsoltan.mirshahi@stago.com (S.M.); genevieve.contantpussard@stago.com (G.C.); 4Laboratoire des Sciences des Procédés et des Matériaux, Centre National de la Recherche Scientifique (UPR 3407), Université Sorbonne Paris Nord, 93430 Villetaneuse, France

**Keywords:** blood viscoelasticity, plasma histones, cancer rheology, clot degradation, cancer thrombosis

## Abstract

**Simple Summary:**

Blood clots are formed when blood vessels are injured. They help stop bleeding and heal wounds, but can also cause serious problems if they block blood flow or break off and travel to other organs. This study investigates how blood clots differ in people with cancer compared to healthy or non-cancerous individuals. This study found that cancer patients have higher levels of histones in their blood, which make their clots stiffer and resistant to lysis. Further, clots formed in cancer patients have higher viscoelastic properties, and hence are harder to break down. These findings suggest that cancer patients have abnormal clotting properties that may increase their risk of developing thrombosis, and evaluating these properties can be helpful in detecting cancer.

**Abstract:**

Background: Blood viscoelasticity and plasma protein levels can play an important role in the diagnosis and prognosis of cancer. However, the role of histones and DNA in modulating blood clot properties remains to be investigated. This study investigates the differences in blood viscoelasticity and plasma protein levels among cancer patients, individuals with other diseases, and healthy individuals. Methods: Blood samples were collected from 101 participants, including 45 cancer patients, 22 healthy individuals, and 34 individuals with other diseases. Rheological properties of clots formed in vitro by reconstituted elements of fibrinogen or plasma were analyzed with an Anton Paar Rheometer, USA. Plasma protein levels of D-dimer, TPA, EPCR, fibrinogen, and histone H3 were measured through ELISA. Blood clots were formed with or without DNA and histones (H3) by adding thrombin and calcium to plasma samples, and were evaluated for viscoelasticity, permeability, and degradation. Results: Cancer patients show higher blood viscoelasticity and plasma D-dimer levels compared to healthy individuals and individuals with other diseases. Our in vitro analysis showed that the addition of histone to the plasma results in a significant decrease in viscoelasticity and mean fiber thickness of the clot formed thereafter. In parallel studies, using plasma from patients, DNA and histones were detected in fibrin clots and were associated with less degradation by t-PA. Moreover, our results show that the presence of DNA and histones not only increases clots’ permeability, but also makes them more prone to degradation. Conclusions: Plasma histones and DNA affect the structure of the clot formed and induce defective fibrinolysis. Moreover, the increased viscoelastic properties of plasma from cancer patients can be used as potential biomarkers in cancer prognosis.

## 1. Introduction

Fibrin clot structure, consisting of a branched fiber network, may directly affect a clot’s fibrinolytic and viscoelastic properties. It is well established that a tight clot is responsible for reduced clot permeability and degradability that may be responsible for a high risk of deep vein thrombosis (DVT) and/or pulmonary embolism (PE) [1,2,3]. In addition, fibrin is a viscoelastic polymer that is under shear stress in the vessel due to blood flow [4,5]. The elasticity (or stiffness) is characterized by a reversible mechanical deformation, whereas the viscosity (or plasticity) is a slow irreversible deformation (creep). The mechanical properties of fibrin are essential for its functions [6,7]. In hemostasis, the clot must form a plug to stop bleeding and this structure must be strong enough to withstand the pressure of arterial blood flow. Fibers are formed when fibrin monomers polymerize spontaneously to form an insoluble multistranded and branched network that entangles the platelet and provides support to form a clot to prevent further bleeding [8]. In the case of a thrombus formation, mechanical properties are also important. If a vessel is partially occluded, the viscoelastic properties of the thrombus will determine whether the flowing blood will cause it to deform reversibly or irreversibly, rupture, or embolize [9].

The viscoelastic properties consist of (1) an elastic component or storage modulus (G′) characterized by a reversible mechanical deformation under shear stress that is used to evaluate the clot stiffness, and (2) an inelastic component or loss modulus (G″) which is a measure of plasticity, that is, slow irreversible deformation. Thus, the storage and loss moduli determine how the clot responds to the forces to which it is subjected, such as flowing blood [10,11].

Epidemiological studies have demonstrated a relationship between myocardial infarction and clot mechanical properties. The in vitro formation of fibrin clots from patients with myocardial infarction shows tight and rigid fibrin network structures compared to controls [12]. Studies have shown clot structure is mainly dependent on fibrinogen concentration [13]. Clots with increased fibrin content show increased stiffness [14]. Clots that are very stiff could also be more friable and have a great tendency to embolize, although almost nothing is known about the relationship between the mechanical properties of fibrin and these pathological properties [6]. It was reported that a modification in the viscoelastic properties of the clot may induce a defective ability of the clot to deform in response to shear stress, contributing to clot embolization [15]. A stiff clot (large G′) deforms less than a softer clot with the same applied stress, and a dramatic increase in G′ is a contributing factor to thromboembolism [5].

Thromboembolism is a serious condition that occurs due to the shifting of part of a clot after breakage. This is a particularly serious concern for cancer patients who have an increased tendency to form abnormal blood clots. One of the major contributors is probably the increased histone levels in cancer patients [16]. Histones have been shown to bind fibrin by FXIIIa, promoting fibrinolytic resistance and hence leading to thromboembolism [17]. Histones are released by dying cells during cancer development and can affect gene expression and genomic stability in cancer cells, as well as blood clotting and thromboembolism in cancer patients.

When compared with deep vein thrombosis (DVT), in vitro clots from patients with pulmonary embolism (PE) exhibit faster lysis times, possibly due to lower fiber density, as well as an earlier establishment of viscoelastic properties, with lag time being significantly faster in PE subjects compared with DVT [18]. Therefore, as already suggested by Weisel [6], it appears that it will be interesting to determine in some patients the stiffness of their in vitro generated clots to evaluate their risk of thromboembolism. Other authors, however, found that clots in recurrent venous thromboembolism patients have a lower elastic modulus than those with nonrecurrent thromboembolism [19].

The relationship between viscoelasticity and thromboembolism is not well understood. Some studies have suggested that stiffer clots are more prone to embolize, while others have found the opposite [19,20,21]. Moreover, the role of histones and DNA, which are released from dying cells during cancer development, in modulating blood clot properties is still unclear. In our study, we evaluated the rheological properties of a clot formed in vitro in peritoneal cancer patients, and the results were compared with those obtained in healthy subjects and in non-cancer patients. In fact, since it was established that the structure of a fibrin clot, characterized in terms of a branched network, directly affects a clot’s fibrinolytic and viscoelastic properties [4], the results of clot stiffness and clot viscosity were compared with those of different parameters of the fibrinolytic pathway and with fibrinogen plasma concentration, since clot stiffness is dependent on fibrinogen concentration [22].

## 2. Material and Methods

### 2.1. Sample Population

This study involved a total of 101 blood samples. Of these, 45 samples were from peritoneal cancer patients (stage IV with metastasis) of different origin (colic: 9; pseudomyxoma: 7; gastric: 6; rectal: 5; ovarian: 6; others: 11), individuals with diseases other than cancer (n = 34), and healthy volunteers (referred to as normal plasma; n = 22). The other diseases group included patients with hypertension, diabetes mellitus, arthritis, anemia, and dyslipidemia. Patients were treated in accordance with the Helsinki Declaration (World Medical Association Declaration of Helsinki. Ethical principles for medical research involving human subjects. Bulletin of the World Health Organization, 2001;79(4):373–374). All the data were anonymously collected, and according to the Loi Jardé (French law amended by Order no. 2016-800 and its implementing decree no. 2016-1537 of 16/11/2016 relating to research involving the human person), no patient consent was needed, as the treatment implemented in this study was the standard recommended therapy.

### 2.2. Samples Tested

Citrated blood: Blood was collected into polypropylene tubes containing 3.2% (0.105 M) sodium citrate (1 part citrate to 9 parts blood dilution). For cancer patients, blood was taken before they underwent routinely planned HIPEC treatment. A part of the citrated blood was used to perform the rheological whole blood clot parameters. The remaining part of the blood was centrifuged at 3000× *g* for 15 min, and 300 μL was used to determine the rheological properties on fresh plasma, while the rest was frozen in 300 μL aliquots. The pooled plasma of healthy individuals was used to evaluate the addition of exogenous substances on clot properties, hereby referred to as pooled plasma.

### 2.3. Interaction of Histones and DNA on Clot Formation

To assess the impact of histones and DNA on clot formation, we purified DNA (350 ng/mL) and used histones (1200 µg/mL) purified from pooled plasma using the Easy-DNA™ Kit (Invitrogen, Carlsbad, CA, USA; K1800-01) and Easy-DNA™ Kit (My BioSource, San Diego, CA, USA; MBS9718630). Purified DNA or histone, or both, in a volume of 30 µL each, was added to 70 µL of pooled plasma. In the case of samples containing only DNA or histone, 30 µL of PBS buffer was added to make up the volume.

### 2.4. Evaluation of Elastic (G′) and Viscous (G″) Modulus

The evaluation of clot elasticity stiffness (G′; storage modulus) and clot viscosity (G″; loss modulus) was performed by adding 48 mU of thrombin in CaCl_2_ to 80 μL of whole blood or fresh or frozen plasma, depending on the group studied. Then, 90 μL of clotting sample was immediately transferred to a rheometer (Anton-Paar) using oscillation at a frequency of 1 Hz and a strain (γ) of 1% for 1500 s. In order to validate the technique, clot elasticity and viscosity tests were performed 20 times over a period of 140 days using the same pool of frozen plasma. From these results, the mean value, standard deviation, variance, and coefficient of variation were calculated. Moreover, the pooled plasma was run at the start of each day when sample viscoelastic parameters were evaluated to validate the rheometer.

### 2.5. Relationship between Clot Elasticity and Fibrin Degradability

Plasma was collected from cancer patients with variable elasticity (G′ value). To 50 μL of plasma, 5 μL of t-PA at 20 μg/mL and 5 μL of thrombin at 10 U/mL in CaCl_2_ 0.25 mM were added. After 5 min, the generated plasmin was neutralized by adding 5 μL of aprotinin at 2.53 IU/mL.

To determine the basal D-dimer level in the sample, a control was performed by performing the same experiment except that thrombin was replaced by 0.15 M NaCl. The clot degradability rate was determined by the difference between D dimers in the clot supernatant (after squeezing the clot) and the basal D dimer level present in the controls, and dividing by basal level. The results are expressed as the percentage of fibrin degraded. The D dimer levels in the samples were evaluated by Sta Compact^®^ ELISA kit (Stago, Asnières sur Seine, IDF, France).

### 2.6. Clot Permeability

Clot permeation was studied as previously described by Okada and Blomback [23]. Pooled plasma samples were recalcified with 25 mM CaCl_2_ (final concentration) and thrombin at a final concentration of 0.45 U/mL. Immediately, 0.5 mL of the mixture was used to fill pre-etched plastic tubes in which one end was sealed with parafilm. The tubes were placed in a moist atmosphere chamber for 2 h to obtain complete coagulation. The tubes were then placed in a holder and connected to a reservoir containing physiologic serum containing 500 μL physiological serum. The time needed for the serum to flow through the plasma clot was measured by noting the time the reservoir was connected and the first drop passed through the clot.

### 2.7. Clot Lysis and Macroscopy of Cancer and Normal Plasma

The clots were prepared using plasma from different cancer patients in 24-well plates by taking 50 μL of plasma and adding 5 μL of thrombin at 10 U/mL in CaCl_2_ 0.25 mM. Once the clot was stabilized, 5 μL of t-PA at 20 μg/mL was added to the clot. To stop the reaction, aprotinin (Bayer) at a volume of 10 μL (2000 U/mL) was added. The clots were then visualized using macro visor assembly for well plates.

### 2.8. Scanning Electron Microscopy (SEM)

The method for scanning electron microscopy was adapted from Iman et al. [24]. Briefly, the blood clots were prepared as described earlier and were fixed using 4% formaldehyde (Merck, Darmstadt, Germany; 104002) for 24 h. After washing three times with adequate quantity of PBS, the clots were then covered with 2.5% glutaraldehyde for 20 min and washed with PBS three times for 5 min each. After PBS, the clots were washed with water to remove any crystals of PBS. The clots were dehydrated with an increasing concentration of ethanol, i.e., 50%, 70%, 90%, and 100%. The clots were then coated with gold nanoparticles to prepare them for scanning electron microscopy using the EMSCD500 apparatus from Leica. The clots were then imaged using an FEG ZEISS ultra 55 scanning electron microscope (Carl Zeiss AG, Oberkochen, Germany). 

### 2.9. Confocal Microscopy

The prepared clots were fixed using 2.5% glutaraldehyde (Merck, Darmstadt, Germany; 111-300-08) for 1 h, washed 3 times with PBS 1X, and dehydrated using an increasing concentration of ethanol. Plasma clots fixed with glutaraldehyde become fluorescent when excited with a 488 nm wavelength and appear green in confocal microscopy. The clots were then stained for H3 histones using primary antibody Histone H3 (Sigma-Aldrich, Burlington, MA, USA, 06-755)^®^ Rabbit mAb and secondary anti-rabbit antibodies conjugated with AF555. The DAPI (Sigma-Aldrich, Burlington, MA, USA, D9542) was used to stain nuclear material, i.e., DNA.

### 2.10. Water Release from Clot

We developed a special technique to measure the water release from the clots using the principle of paper chromatography. The plasma clots were prepared using the plasma of cancer patients with different elasticity parameters. Whatman filter paper, vertically hung, was placed in contact with the clots. This causes the water in plasma clots to move upward through the Whatman filter paper. The height travelled by the water was marked at regular intervals of 15 s for 1 min, and then 30 s for 5 min.

### 2.11. Clot Thickness Measurements

The SEM images from clots were analyzed using Fiji to measure thickness and the length of individual fibers or fiber bundles [25]. The Fiji scale was calibrated using the scale bars provided in the SEM images from the microscope. The mean fiber thickness was taken by measuring fibers in different images taken from different clots with similar viscoelastic properties, i.e., elasticity and viscosity modulus.

### 2.12. Biological Parameters

The samples were evaluated for the concentrations of D-dimer, fibrinogen, PAI-1, t-PA, EPCR, heparinase, and tissue factor. The tests were performed using the following commercial kits: D-Di (STA-Liatest D-Di Plus, Stago, Asnières sur Seine, IDF, France), fibrinogen level (STA-Liquid Fib, Stago, Asnières sur Seine, IDF, France), PAI-1 (Asserachrom PAI-1, Stago, Asnières sur Seine, IDF, France), t-PA (Asserachrom t-PA, Stago, Asnières sur Seine, IDF, France), sEPCR (Asserachrom s-EPCR, Stago, Asnières sur Seine, IDF, France), heparanase (Insight Biotechnology Limited, Middlesex, UK), Histone H3 (Bio-Techne R&D Systems, Minneapolis, MN, USA) and tissue factor (Tissue Factor Quantikine ELISA, R&D, Paris, France).

### 2.13. Statistical Analysis

The statistical analysis reported was performed using SPSS version 26. The groups were compared using a t test for 2 groups or the post hoc Sidak test for more than 3 groups. The correlation reported is the Spearman correlation for a non-normal distribution. The values are reported as the mean ± S.D.

## 3. Results

### 3.1. Demographics

Our study included a total of 101 patient samples, with 34 individuals with other diseases, 45 cancer patients, and 22 healthy individuals, as described in Table 1.

### 3.2. Plasma Proteins

A total of five plasma proteins were compared for differences in levels among healthy, other disease, and cancer patients. The only statistically significant difference was found for D-dimer, where cancer samples had significantly higher D-dimer levels than other disease samples (1.81 ± 0.15 vs. 0.83 ± 0.17), as well as normal samples (1.81 ± 0.15 vs. 0.35 ± 0.02), with *p* values < 0.0001 (Figure 1). Although fibrinogen and EPCR levels were increased in other disease and cancer samples compared to normal samples, the difference was not significant.

### 3.3. Effect of DNA–Histone on Clot Formation

We performed an in vitro experiment to evaluate the impact of DNA and histones on clot structure. The clots were prepared from plasma pooled using healthy individuals and adding DNA, histones, or both to the plasma. The clots were then analyzed by scanning electron microscopy (SEM), as shown in Figure 2. The addition of DNA results in decreased mean fiber thickness compared to clots formed without added DNA (140 vs. 110 nm) (Figure 2B). Adding histones along with DNA did not further cause any decrease in fiber thickness (Figure 2C); however, histones in the absence of DNA resulted in the lowest mean fiber thickness (60 nm) (Figure 2D). A normal plasmatic clot is presented in Figure 2A. Interestingly, the addition of nucleosome elements in plasma modified the porosity of fibrin networks (Figure 2C-D). Interestingly, adding DNA to clots caused increased thickness of the fiber knots (bundles) formed in the clot, i.e., the mean thickness of the bundle increased to 350 nm from 310 nm. However, adding histones significantly reduced the mean bundle thickness to 150 nm. The results of the thickness and bundle measurements and their nodes are presented in Figure 2E–G, respectively.

To assess the effect of DNA and histones on clot permeability, we prepared a clot in a capillary tube to clog it properly and then added water to the top of the tube. The time required for the water to pass through the clot can be taken as an inverse function of clot permeability, i.e., the more time needed, the lower the permeability will be. Our results show that histones were more potent in increasing clot permeability as the time was decreased to 33 ± 4 min, compared to 57 ± 5 min for normal clots. The addition of DNA decreased the time to 45 ± 3 min. However, when histone was added along with DNA, the time was further decreased to 38 ± 3 min (Figure 2G).

### 3.4. Histone Levels in Cancer Patients

We evaluated the cancer samples for histone presence. The histone levels ranged from 0.15 ng/mL to 91.62 ng/mL, with a mean of 9.53 ± 2.88 ng/mL. Although we did not have a big enough sample size to evaluate the statistical significance, appendicular adenocarcinoma samples showed higher histone levels, with a mean level of 40.98 ± 28.01 ng/mL, which is almost twice as high as that of the second highest cancer type, i.e., pseudomyxoma, with mean levels of 21.08 ± 11.14 ng/mL (Figure 3).

### 3.5. In Vitro and In Vivo Fibrinogen Role in Viscoelasticity

To evaluate how fibrinogen levels affect the viscoelastic properties of clots and their permeation, we prepared clots with additional exogenous fibrinogen (Stago, Asnières sur Seine, IDF, France) at concentrations of 3.4, 7, and 9 mg/mL. Our results show that the viscoelasticity of clots increases with increasing concentrations of fibrinogen(Figure 4).

Moreover, this was confirmed ex vivo using the samples from healthy individuals, individuals with other diseases, and cancer patients, as there was a strong positive correlation of fibrinogen levels in plasma with fresh blood elasticity (r_s_ = 0.305), fresh blood viscosity (r_s_ = 0.436), fresh plasma elasticity (r_s_ = 0.446), fresh plasma viscosity (r_s_ = 0.340), frozen plasma elasticity (r_s_ = 0.621), and frozen plasma viscosity (r_s_ = 0.396).

### 3.6. Fibrin Clots from Carcinomatosis Plasma Contain Nucleosome Materials That Colocalize with Fibrin Networks

The colocalization of fibrin with DNA is shown in Figure 5. The figure shows the components and structure of a clot in a cancer patient. Fibrin (green) associated with nucleotides (blue) was identified (Figure 5B). The fibrin networks from normal plasma (Figure 5A) and cancer patient plasma (Figure 5B) identified via confocal fluorescence microscopy show the structural modification of the arrangement of the fiber in the clots. In this experiment, the fibrin fixed by glutaraldehyde under the microscope via ultraviolet light took on a green color. Nucleic materials were stained by DAPI. The partial and complete clot lyses of a carcinomatosis patient are presented in Figure 5C,D, respectively. Figure 5E presents the nucleosome content of clots after total fibrin fiber lysis. The exogenous DNA added to the plasma is presented in Figure 5F. Normal plasma lysis by t-PA is presented in Figure 5G. These results are in favor of DNA colocalization with fibrin networks.

In another experiment, we detected histone materials (H-3) associated with fibrin networks. The results presented in Figure 6 show the fibrin filament in green (Figure 6B) associated with histone H3 labeled in red (Figure 6A) by the anti-H3 antibody D1H2. The combination of the two proteins is shown in Figure 6C. The graph shows the quantifications of fibrin and histone H3 in green and red, respectively, depending on the area studied (Figure 6C), which makes it possible to identify the colocalization of the two compounds within the clot.

### 3.7. Reproducibility of the Viscoelasticity Measurement Technique

The intra-assay reproducibility of G′ and that of G″ were determined using the same frozen pool of normal plasma that was frozen in fractions of 300 µL. The day-to-day reproducibility of G′ and that of G″ were also determined for 140 days using a fresh aliquot of the frozen sample. The mean value +/− SD, the coefficients of variation (CV), and the variance were determined (results not shown). Our technique showed %C.V. < 10 for elasticity as well as viscosity, showing good reproducibility of the technique.

### 3.8. Impact of Elasticity on Clot Degradation and Fiber Thickness

To assess how elasticity affects clot structure and degradation, we evaluated the degradation rate of different cancer samples with increasing elasticity. The degradation rate had a strong negative correlation (r_s_ = −0.846) with elasticity and the initial D-dimer levels (r_s_ = −0.35) present in plasma (Figure 7A,B).

Moreover, SEM analysis showed that the clots (*n* = 5 each) formed with higher elasticity had significantly thick fiber formation compared to those with lower elasticity with a mean thickness of 154 ± 22 nm, as compared to 91 ± 31 mm (*p* < 0.0001). Moreover, we evaluated the water release from the clot using the special technique of paper chromatography by touching the clot with filter paper and measuring the distance traveled by the water at specific intervals. We found that clots formed with plasma with higher elasticity released less water compared to the ones with lower elasticity. This was further confirmed through trypan blue release, as less dye was released from clots of higher elasticity in comparison to those formed from lower elasticity.

### 3.9. Fibrin Formed from the Plasma of Patients with Carcinomatosis Presents Lytic Resistance to t-PA

First, we demonstrated in vitro the visualization of clot degradation. The clots that are sensitive to the presence of the fibrinolytic enzyme t-PA (fibrin formed from control plasma) and which fibrinolysis is defective (fibrin formed with carcinomatosis patient) are presented successively in Figure 8A. In this experiment, the supernatants of clots degraded by t-PA for control and all carcinomatosis patients were collected, and a D-dimer test was performed for the evaluation of defective fibrinolysis (Figure 8B). Figure 8C shows some examples of sensitivity to t-PA of fibrins formed from the plasma of several patients (the map of the plates is presented in the table according to each cancer). We found that clots of cancer patients were degraded much less (on average 30%) than normal pooled plasma. Moreover, there is little variation depending on the cancer subtypes, with pseudomyxoma patients showing the lowest degradation (14% to that of normal plasma) compared to gastric patients, who showed more degradation compared to pseudomyxoma, but still quite a lot less than normal pooled plasma, i.e., 40% of the degradation of pooled plasma.

### 3.10. Elasticity and Viscosity of Cancer and Other Samples

Our results suggest that fresh cancer blood samples had significantly higher elasticity (114.17 ± 12.45) compared to other disease (72.78 ± 7.56; *p* value = 0.008) and normal samples (70.90 ± 8.50; *p* value= 0.017). Moreover, when compared for fresh plasma, cancer samples had significantly higher elasticity (127.41 ± 25.72) compared to other disease samples (75.00 ± 5.57; *p* value = 0.040). However, there was no significant difference between normal and cancer or other disease samples. Significantly increased elasticity was also observed for cancer frozen plasma samples (69.55 ± 7.79) compared to other disease (47.03 ± 3.34; *p* value = 0.035) and normal frozen plasma samples (39.31 ± 5.03; *p* value = 0.014).

Similar to elasticity, we found that cancer fresh blood samples showed significantly higher viscosity than other disease and normal fresh blood samples. However, no significant difference was found among the three groups for the viscosity of fresh plasma samples or frozen plasma samples (Figure 9). Moreover, there was no significant difference between the viscoelastic properties of plasma obtained from healthy individuals or individuals with other diseases.

### 3.11. Elasticity and Viscosity per Gram of Fibrinogen

We found that the ratio of elasticity to fibrinogen was much higher in cancer samples than in healthy individuals and individuals with other diseases (Figure 10). The elasticity per gram for fresh blood samples was higher in individuals with cancer (39.22 ± 3.76) compared to individuals with other diseases (21.57 ± 2.35; *p* value < 0.0001). However, it was not significant compared to normal samples (27.67 ± 3.77; *p* value= 0.067). Similarly, for fresh blood samples, the mean value of elasticity per gram of fibrinogen was significantly higher in cancer samples (41.56 ± 6.73) compared to individuals with other diseases (23.09 ± 2.07; *p* value = 0.01), but not compared to normal individuals (28.09 ± 4.9; *p* value = 0.16). However, in frozen plasma samples, the elasticity per gram of fibrinogen was significantly higher in cancer patients (20.19 ± 1.55) compared to individuals with other diseases (13.53 ± 0.83; *p* value = 0.002) and healthy individuals (13.15 ± 1.52; *p* value = 0.006). Interestingly, we found that significantly higher viscosity was only found in the blood samples of cancer patients (2.80 ± 0.27) compared to other disease (1.68 ± 0.15; *p* value = 0.001) and normal samples (1.83 ± 0.21; *p* value = 0.014). However, no significant difference was found among the groups for fresh or frozen plasma samples.

## 4. Discussion

Blood clot properties are affected by various factors such as altered plasma protein levels, fibrinogen, D-dimers, histones (H3), fibrin structure, and shear stress due to environmental factors or diseases, such as cancer [13,17,26,27]. In cancer patients, blood clotting is often dysregulated and associated with an increased risk of thrombosis and inflammation. Cancer patients have an increased risk of developing deep vein thrombosis and pulmonary embolism [28]. Cancer cells themselves can activate coagulation pathways by releasing cytokines, tissue factors, etc. Histones have been shown to inhibit fibrinolysis through linking to fibrin [17]. However, the role of histones and DNA, which are released from dying cells and tumors, in modulating blood clot properties is not well understood in the context of cancer. In this study, we investigated the differences in blood viscoelasticity and plasma protein levels among the fresh blood and plasma and frozen plasma of cancer patients, individuals with other diseases, and healthy individuals. We also examined the impact of the incorporation of histones and DNA to blood clots on their viscoelasticity, permeability, and degradation. Our findings provide new insights into the implications of blood clot formation and lysis in cancer patients.

We found that cancer patients had significantly higher levels of D-dimer, with no difference in the levels of fibrinogen, PAI, TPA, or EPCR. D-dimer, a marker of fibrin degradation, suggests that cancer patients have increased active clotting and lysis. These high levels of D-dimer may lead to an increased risk of thrombosis and bleeding. The findings regarding D-dimer levels are consistent with previous studies [29]. However, in contrast to other studies, we did not find altered levels of fibrinogen, PAI, TPA, or EPCR in cancer patients. The discrepancy might be attributed to the low number of patients or different types of cancer studied, as our sample size was limited to stage IV cancer patients.

Furthermore, adding DNA or histones had variable effects on clot properties such as fiber thickness, bundle thickness, and elasticity. DNA resulted in decreased fiber thickness but did not affect bundle thickness, while the addition of histones resulted in a decreased thickness of individual fibers as well as bundles. The effect of histones was reduced when added along with DNA, which might be attributed to the formation of complexes of DNA and histones, resulting in a reduced effect of histones on fiber formation. Interestingly, these results are in contrast to previous studies that show that the addition of DNA or histones to clot results in thicker fibrin formation [30]. Based on our observations, DNA filaments and histones can be associated with fibrin during clot formation and can generate bundles. This has two impacts on the clot. First, it increases the permeability as the interfibrillar pores are modified and pass more liquids. Second, it modifies the accessibility of fibrinolytic enzymes such as plasmin to fibers. Therefore, even if it degrades the fibrin associated with the DNA filament or the histone, the architectural structure of the clot can still persist, making the clot less prone to degradation. We found that histone levels vary differently among cancer patients, and even for stage IV, the levels of histones vary across cancer origins. However, these results need to be confirmed in a larger cohort. In our study, cancer patients of appendicular adenocarcinoma and pseudomyxoma origin showed higher levels of histones, while lower levels were observed for breast cancer, colic cancer, and ovarian cancer patients. These histone levels may reflect the extent of cell death or tumor burden in cancer patients and may have implications for their risk of thrombotic and hemorrhagic complications. However, the sample size is quite small, which limits the generalization of these findings.

To evaluate the viscoelastic parameters of blood clots and compare them among healthy individuals, individuals with other diseases, and cancer patients, we developed an in-house technique using an Anton-Paar rheometer. We validated our technique by using a frozen pool of plasma run every time before sample analysis. The % C.V < 10 showed that our method was quite robust and repeatable for evaluating the viscoelastic parameters of blood clots. When we tested the samples from cancer patients, patients with diseases other than cancer, and healthy individuals, we found that cancer patients showed significantly higher viscoelastic properties compared to individuals with other diseases or healthy individuals. The elastic modulus (G′) was more consistent in providing this finding, especially when evaluated using fresh blood or frozen plasma samples. Moreover, we found that the viscoelastic properties of fresh plasma were different from those of fresh blood, probably due to the presence of cells. When plasma was frozen at −22 °C, we found a drastic decrease in both elastic and viscosity moduli, irrespective of sample type, i.e., normal, other disease, or cancer samples. Furthermore, repeated refreezing cycles cause a decrease in the viscoelastic properties of plasma (results not shown); therefore, aliquots are needed to preserve blood samples for rheological evaluation while avoiding repeated freeze–thaw cycles. This suggests the alteration of protein structure during the process of freezing. Our in vitro studies show that clots with higher elasticity, i.e., from cancer patients, have thick and condensed fibers and are less prone to degradation than clots formed in blood with lower elasticity. Therefore, clots of cancer patients with high G′ values have low fibrin degradability, which could explain why high G′ clots might be responsible for the high risk of venous thromboembolism.

No accelerated establishment of viscoelastic properties, documented by a decrease in lag time and an increase in the rate of viscoelastic property formation, was observed in the cancer patients tested. However, at the time of the evaluation of the viscoelastic properties of the clot, no patient presented deep vein thrombosis (DVT) or pulmonary embolism (PE). In fact, Martinez et al. reported that, compared with DVT clots derived from PE subjects, clots derived from PE subjects showed an accelerated establishment of viscoelastic properties [18].

Despite the limited sample size of our study, we present a novel finding on how histones and DNA modulate the properties of blood clots in cancer patients. Our findings suggest that measuring the viscoelastic parameters of blood clots, along with the plasma protein levels and the histone and DNA content, could be useful in assessing the hemostatic status and thrombotic risk of cancer patients.

## 5. Conclusions

In conclusion, this study showed that cancer patients have altered blood clotting properties compared to individuals with other diseases and healthy individuals. We found that cancer patients had higher plasma histone levels, higher clot elasticity and viscosity, and an increase in viscoelastic properties independent of fibrinogen levels. These findings suggest that cancer patients have a hypercoagulable state that may affect their hemostatic balance and thrombotic risk. The mechanisms underlying these alterations are not clear and may involve the interaction between the DNA and histone network associated with fibrin and cancer cells, host cells, and coagulation factors. Further studies are needed to elucidate the safety of circulating DNA and histones, the molecular and cellular basis of these changes, and their clinical implications for the defective lysis of cancer-associated thrombosis.

## Figures and Tables

**Figure 1 cancers-16-00928-f001:**
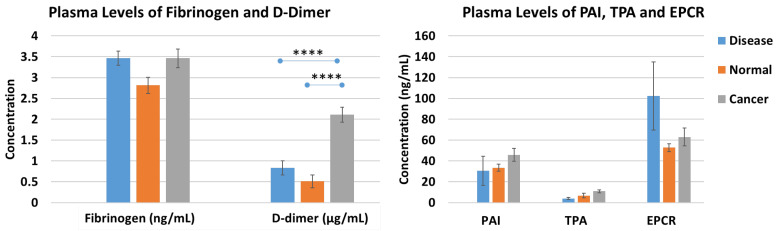
Levels of fibrinogen, D-dimer (left), PAI, TPA, and EPCR (right) in plasma samples of cancer patients, individuals with other diseases, and healthy individuals. Error bars represent the mean ± S.E. (**** *p* value < 0.0001; Sidak test).

**Figure 2 cancers-16-00928-f002:**
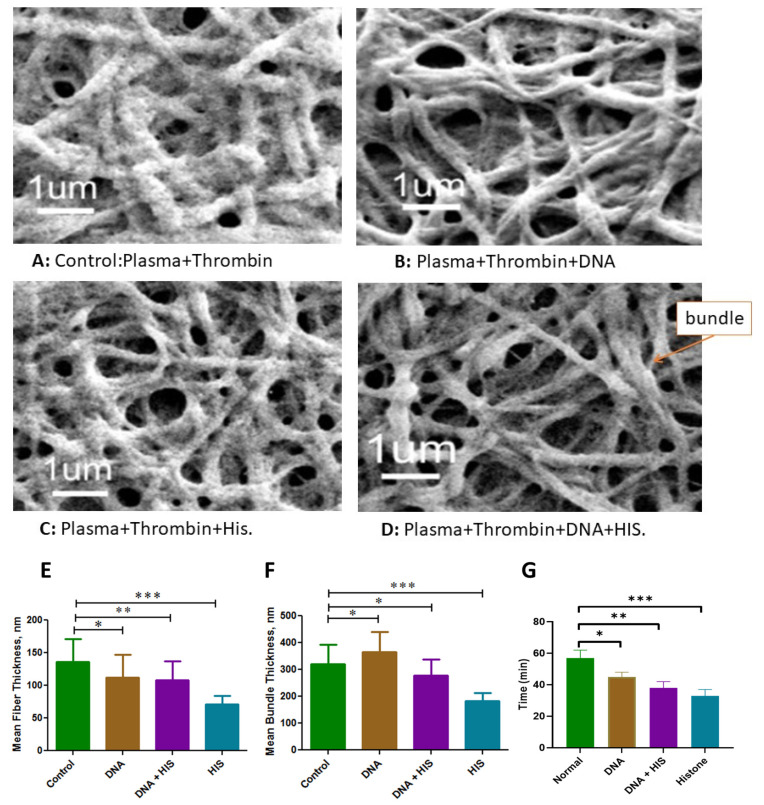
Effect of DNA and histones on the fibers of formed clots using pooled plasma from healthy individuals (n = 10). (**A**) Clot formed without DNA and histones. (**B**) Clot with DNA. (**C**) Clot with histones. (**D**) Clot with DNA and histones. The bundle formations are the network of multiple fibrins interlinking together. (**E**) Mean fiber thickness in the clot formed with or without added DNA and histones. (**F**) Mean bundle thickness in the clot with or without added DNA and histones. (**G**) Impact of DNA and histones on clot permeability. The bars represent the time taken by the water to cross the clot formed with or without added DNA or histones, or both. Less time required means more clot permeability. Error bars present the mean ± S.E. (* *p* value < 0.05, ** *p* value < 0.01, *** *p* value < 0.001).

**Figure 3 cancers-16-00928-f003:**
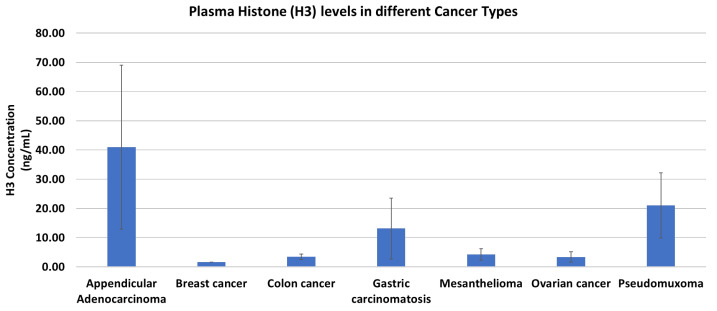
Histone (H3) levels among different cancer types. Appendicular adenocarcinoma patients show the highest histone (H3) levels; however, there is variability among patients of the same cancer type. Error bars represent mean ± S.E.

**Figure 4 cancers-16-00928-f004:**
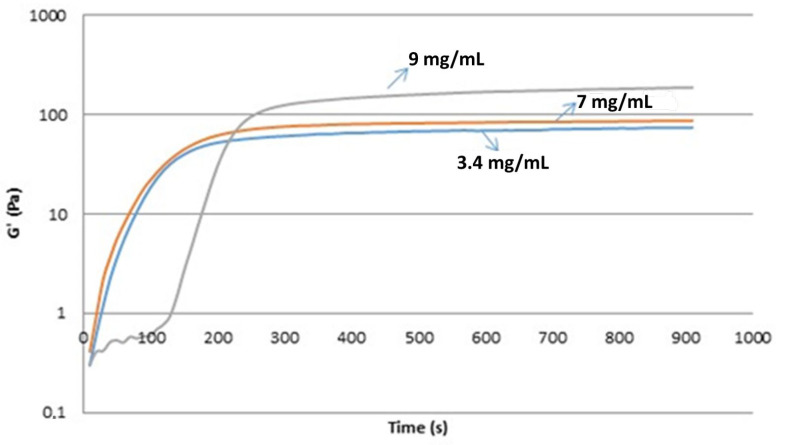
Effect of in vitro fibrinogen levels on the elasticity of the clot. The elasticity increases with increasing concentrations of fibrinogen.

**Figure 5 cancers-16-00928-f005:**
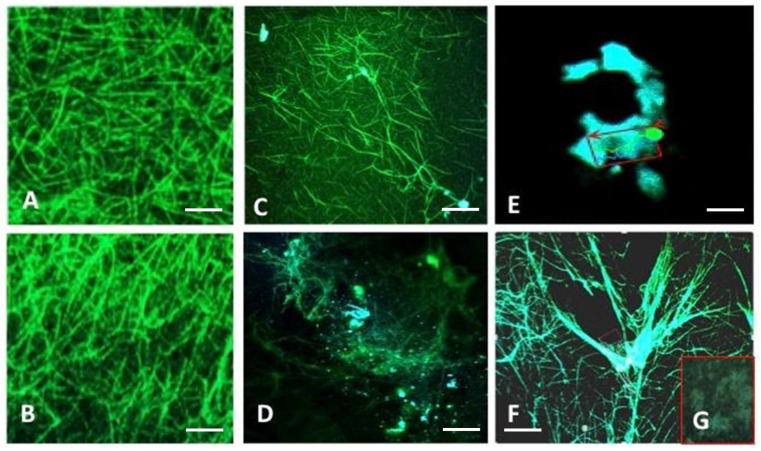
Colocalization of fibrin (green) and DAPI (blue) through confocal microscopy (*n* = 10). (**A**) Fibrin clot of normal plasma. (**B**) Fibrin clot from a cancer patient. (**C**) Partial lysis of clot from cancer patient by exogenous t-PA. (**D**). Presence of DNA and fibrin residue after complete lysis of clot by exogenous t-PA. (**E**). Magnified nucleosomal content residue left after the complete lysis of clot from cancer patient. (**F**). Colocalization of exogenous DNA with fibrins. Scale bar (**A**–**F**) represents 20 µm.(**G**). 10x magnification of red box in F. Showing complete lysis of normal plasma clot.

**Figure 6 cancers-16-00928-f006:**
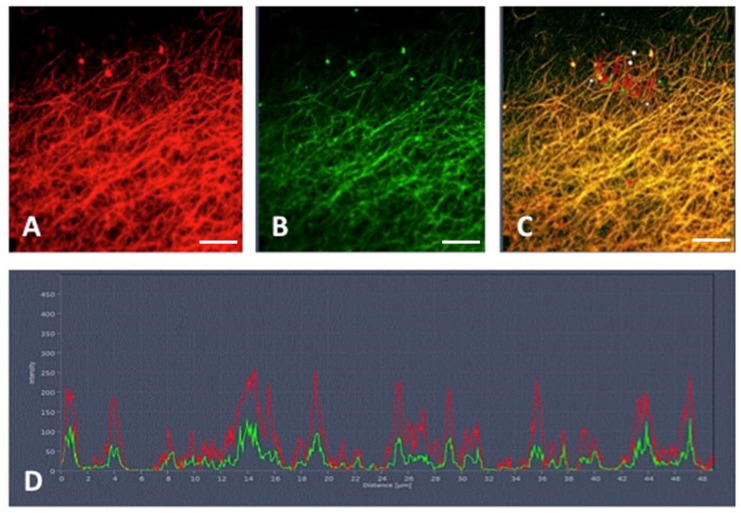
Colocalization of (**A**) histone H3 (red) with (**B**) fibrin (green). (**C**) Merged image representing the colocalization for fibrin and histones in plasma clot of cancer patients. (**D**) Histogram showing the overlap of fibrin and histone channels (*n* = 10). Scale bar represents 20 µm.

**Figure 7 cancers-16-00928-f007:**
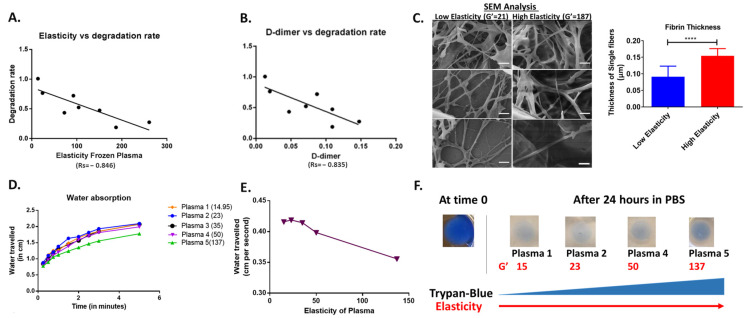
Impact of elasticity on clot degradation. Correlation of plasma degradation rate with elasticity modulus (**A**) and D-dimer levels (**B**) for different plasma samples (*n* = 8). (**C**) SEM scan of clots formed using plasma with lower elasticity (left) and higher elasticity (right). Scale bar represents 1 μm. The graph shows the composite thickness of fibrin in clots formed with low or high elasticity. Error bars represent mean ± S.E. (**** *p* value < 0.0001; Mann-whitney test) (**D**). Release of water by clots (*n* = 5) formed using varying levels of plasma elasticity (elastic modulus, G′ shown in brackets). (**E**) The correlation between water release and the elasticity of plasma clots (*n* = 5). (**F**) Release of water-soluble trypan blue dye from clots (*n* = 5) prepared from plasma of increasing elasticity. The plasma clot at time 0 is shown on the left in dark blue.

**Figure 8 cancers-16-00928-f008:**
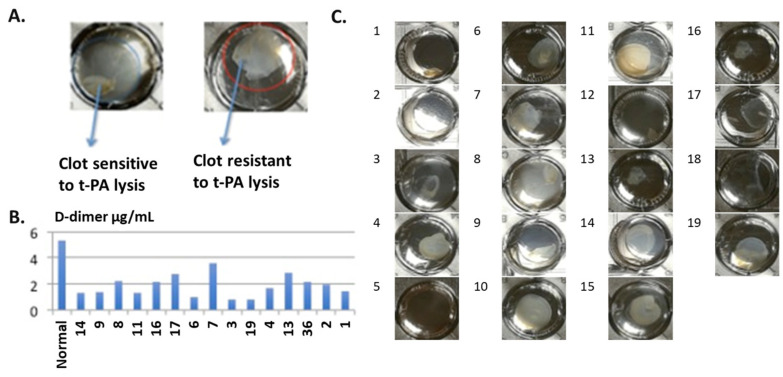
Clot lysis by t-PA. (**A**) Clot sensitive to t-PA lysis (left) and resistant to t-PA lysis (right) after 24 h. (**B**) D-dimer levels in supernatant after 24 h of t-PA activity. Lower levels of D-dimer indicate decreased clot lysis. (**C**) Clot image after 24 h of t-PA activity through a macro visor in a 24 h well plate. The image numbers correspond to the numbers in (**B**). 1–6, ovarian cancer; 7–10, pseudomyxoma; 11–13, colon cancer; 14–16, mesothelioma; 17, breast cancer; 18, appendicular carcinoma; 19, not confirmed.

**Figure 9 cancers-16-00928-f009:**
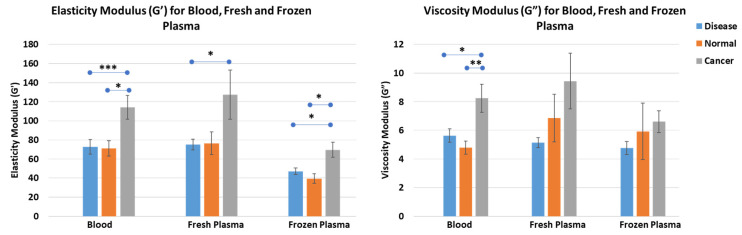
Elasticity (G′; **left**) and viscosity (G″; **right**) moduli of clots formed from the plasma of other disease, cancer, and normal samples. Cancer samples show significantly higher elasticity for blood, fresh, and frozen plasma compared to other disease and normal samples. However, no difference was observed for the viscosity among the cancer samples compared to other disease and normal samples. Error bars represent mean ± S.E. (* *p* value < 0.05, ** *p* value < 0.01, *** *p* value < 0.001).

**Figure 10 cancers-16-00928-f010:**
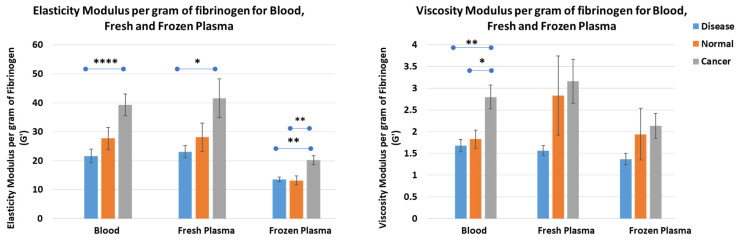
Elasticity (G′; **left**) and viscosity (G″; **right**) per gram of fibrinogen for clots prepared using other disease, normal, and cancer plasma. A significantly higher electricity modulus per gram of fibrinogen was found for cancer samples compared to individuals with other diseases for blood, fresh plasma, and frozen plasma. However, compared to normal samples, the difference was observed only in frozen plasma samples. A significantly higher viscosity per gram of fibrinogen was found in frozen plasma of cancer samples compared to other disease and normal samples. Error bars represent mean ± S.E. (* *p* value < 0.05, ** *p* value < 0.01, **** *p* value < 0.001).

**Table 1 cancers-16-00928-t001:** Demographics table for the collected blood samples.

		Disease (*n*= 34)	Normal (*n* = 22)	Cancer (*n* = 45)
		*n* (%)	*n* (%)	*n* (%)
Age Groups	Not Reported	1 (2.94)	4 (18.18)	2 (4.44)
	Under 18	1 (2.94)	-	-
	18–25	-	1 (4.55)	-
	26–39	3 (8.82)	5 (22.73)	3 (6.67)
	40–55	9 (26.47)	3 (13.64)	15 (33.33)
	Above 55	20 (58.82)	9 (40.91)	25 (55.56)
Gender	Not Reported	1 (7.7)	5 (38.5)	7 (53.8)
	Female	18 (40.8)	8 (16.3)	21 (42.9)
	Male	15 (38.5)	9 (23.1)	15 (38.5)
Disease	Diabetes	10 (9.9)	-	-
	Hypertension	6 (5.9)	-	-
	Other Disease (Arthritis, Anemia, and dyslipidemia)	18 (17.8)	-	-
Cancer type	Appendicular Adenocarcinoma	-	-	3 (6.67)
	Breast Cancer	-	-	1 (2.22)
	Colon Cancer	-	-	13 (28.89)
	Gastric Carcinomatosis	-	-	6 (13.33)
	Mesothelioma	-	-	5 (11.11)
	Ovarian Cancer	-	-	9 (20.00)
	Pseudomyxoma	-	-	8 (17.78)

## Data Availability

Dataset available on request from the authors.

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
