# Peer review of "Blood Clot Dynamics and Fibrinolysis Impairment in Cancer: The Role of Plasma Histones and DNA"

_cancers, 2024, doi:10.3390/cancers16050928_

Round 1
Reviewer 1 Report (Previous Reviewer 1)
Comments and Suggestions for Authors
All my previously raised comments have been addressed in the revised version of the text
Author Response
We would like to thank the reviewer for his time taken to review our manuscript and helping us improve it.
Regards
Reviewer 2 Report (Previous Reviewer 2)
Comments and Suggestions for Authors
The authors have now revised their manuscript but have not responded to my comments. The authors should correct according to the previous recommendation
Pages 1-2: The authors only briefly state the function of fibrin in hemostasis, it is necessary to state this function in more detail. ,, Fibrin monomers spontaneously polymerase, resulting in the formation of thicker fibers and an insoluble multistranded and branched fiber network that entangles platelets to form a blood clot, blocking the damaged blood vessel and preventing further bleeding“. These statements have been published in a manuscript, which the authors should cite. J. Clin. Med. 2022, 11(4), 1083; https://doi.org/10.3390/jcm11041083
Page 2: The authors should refer in this section to a more recent manuscript where it was described that the concentration of fibrinogen is the main determinant of the structure (fiber thickness, branching and density of the network) of the resulting fibrin network: Biomedicines 2020, 8(12), 605; https://doi.org/10.3390/biomedicines8120605
Although there are 21 references in the manuscript, only a small number are references from the last 5 years. The authors should also add more recent references
Author Response
Dear Reviewer,
We would like to apologize for your some of your comments being overlooked.
The two references are really interesting and related to our study and has been added to line 52-54 & 66-67 as follows:
1) Fibers are formed when fibrin monomers polymerize spontaneously to form an insoluble multistranded and branched network that entangles the platelet and provide support to form a clot to prevent further bleeding [8, Ref: J. Clin. Med. 2022, 11(4), 1083; https://doi.org/10.3390/jcm11041083]
2) Studies have shown clot structure is mainly dependent on fibrinogen concentration [13, Ref: Biomedicines 2020, 8(12), 605; https://doi.org/10.3390/biomedicines8120605].
Moreover, further references to latest articles (After 2019) have been added to support the statements in the paper.
Please find attached the updated script. Thank you for your time.

Reviewer 3 Report (Previous Reviewer 3)
Comments and Suggestions for Authors
The authors have done considerable work revising the manuscript and I mostly have minor comments.
However, the number measurements/readings (n) must be provided in each figure legend, including a mention of what the data/bars represent (i.e mean +SE). also, please state how many clots were imaged and what n is in fig 2, 5, 6, and 7.
- lines 265-268 seems out of place.
- line 276, a reference to panel G should be added to Figure 2
- line 279, Figure 3 should be moved to the end of line 285 and replaced by "data not shown".
- the ELISA kit used to measure histones has not been added to 2.12
Round 2
Reviewer 2 Report (Previous Reviewer 2)
Comments and Suggestions for Authors
The presented manuscript has been corrected in response to the suggestions. The authors have followed the recommendations of the reviewer. After the revision, the provided data and addition of the results became more clear. I would like to thank the authors for resubmitting the manuscript and explaining the obscure points from the previous version.
This manuscript is a resubmission of an earlier submission. The following is a list of the peer review reports and author responses from that submission.
Round 1
Reviewer 1 Report
Comments and Suggestions for Authors
The research manuscript by Ullah et al “Blood plasma nucleosomes’ ability to alter fibrin structure and fibrinolysis in cancer patients” examines several characteristics of the blood clot in healthy individuals and those with cancer or other diseases (both in fresh blood and frozen specimens). They also examine the impact of histones and DNA on aforementioned characteristics of plasma and blood clot.
They find that histones and DNA alter such characteristics (their addition resulted in significant decrease in viscoelasticity and mean fiber thickness of the clot, and also made them prone to degradation), and that blood clots from cancer patients are different compared to healthy individuals; moreover, cancer patients had higher blood viscoelasticity and plasma D-dimer levels. The study cohort of circa 100 participants is relatively small, but is big enough for a pilot study.
Please find my comments and suggestions below.
The title of the article is “too broad”.
I suggest replacing it with something like “Blood plasma nucleosomes’ ability to alter fibrin structure and fibrinolysis in patients with some types of solid tumours”
I suggest replacing Line 15 “blood clots w.r.t viscoelasticity, permeability, and degradation” with “blood clots with reference to viscoelasticity, permeability, and degradation”.
Figure 3, Figure 7, Figure 9, and Figure 10. The text size is too small and almost illegible.
Please use uniform wording throughout the text (for example, always say “D-dimer”, do not use “d-dimer” wording in the same text).
The aforementioned structural and bio-mechanical characteristics of the clot are most likely different in individuals with different physiological and pathological states (as demonstrated in this work), supposedly they might be affected by diet, medication, gender, circadian rhythms, therapy such as chemotherapy or radiation therapy, etc. Please discuss in detail and refer to studies supporting (or challenging) this statement. The authors discussed the properties of the clots in TE and some other cardiovascular diseases, but I suggest elaborating on writing and overview other conditions as described above. What about clots in case of the “blood cancers”? What about long COVID-19 as a co-morbidity, as it's known that long COVID-19 leads to the formation of the “micro clots” and aberrant protein aggregation in the bloodstream?
What about cancer medications affecting blood clot formation?
What is the immediate clinical value of the finding?
Reviewer 2 Report
Comments and Suggestions for Authors
The authors describe a study focused blood plasma nucleosomes' ability to alter fibrin structure and fibrinolysis in cancer patients. The authors found interesting results that cancer patients, especially those with pseudomyxoma, showed higher blood viscoelasticity and plasma D-dimer levels than normal and diseased subjects.
This is an interesting manuscript, however, some parts need to be corrected as recommended.
Page 1: The authors only briefly state the function of fibrin in hemostasis, it is necessary to state this function in more detail. ,, Fibrin monomers spontaneously polymerase, resulting in the formation of thicker fibers and an insoluble multistranded and branched fiber network that entangles platelets to form a blood clot, blocking the damaged blood vessel and preventing further bleeding“. These statements have been published in a manuscript, which the authors should cite. J. Clin. Med. 2022, 11(4), 1083; https://doi.org/10.3390/jcm11041083
Page 2: lines 88-93 The authors should refer in this section to a more recent manuscript where it was described that the concentration of fibrinogen is the main determinant of the structure (fiber thickness, branching and density of the network) of the resulting fibrin network: Biomedicines 2020, 8(12), 605; https://doi.org/10.3390/biomedicines8120605
Table and figures are processed correctly.
The authors list only 13 references, of which only 3 are references from the last 5 years. New references should be added in the manuscript.
Reviewer 3 Report
Comments and Suggestions for Authors

Comments on the Quality of English Language
See evaluation report